# High-Resolution Strain/Stress Measurements by Three-Axis Neutron Diffractometer

**DOI:** 10.3390/ma13235449

**Published:** 2020-11-30

**Authors:** Pavol Mikula, Vasyl Ryukhtin, Jan Šaroun, Pavel Strunz

**Affiliations:** Department of Neutron Physics, Nuclear Physics Institute ASCR, v.v.i., 250 68 Řež, Czech Republic; ryukhtin@ujf.cas.cz (V.R.); saroun@ujf.cas.cz (J.Š.); strunz@ujf.cas.cz (P.S.)

**Keywords:** residual stresses, neutron diffraction, three axis setting, high resolution, bent crystal monochromator, bent crystal analyzer

## Abstract

Resolution properties of the unconventional high-resolution neutron diffraction three-axis setup for strain/stress measurements of large bulk polycrystalline samples are presented. Contrary to the conventional two-axis setups, in this case, the strain measurement on a sample situated on the second axis is carried out by rocking the bent perfect crystal (BPC) analyzer situated on the third axis of the diffractometer. Thus, the so-called rocking curve provides the sample diffraction profile. The neutron signal coming from the analyzer is registered by a point detector. This new setup provides a considerably higher resolution (at least by a factor of 5), which however, requires a much longer measurement time. The high-resolution neutron diffraction setting can be effectively used, namely, for bulk gauge volumes up to several cubic centimeters, and for plastic deformation studies on the basis of the analysis of diffraction line profiles, thus providing average values of microstructure characteristics over the irradiated gauge volume.

## 1. Introduction

Residual stresses are typical phenomena associated, e.g., with the welding of different kinds of structural material. Stresses are generally responsible for the deformation of structures during production and subsequently influencing the behavior of these structures during service [1,2,3]. However, they also occur when external load or some kind of shape forming is applied to the sample. X-ray diffraction and neutron diffraction are excellent nondestructive techniques for the determination of strain/stress fields in polycrystalline materials. X-ray diffraction, which due to a strong attenuation of X-rays in the material, is limited to sample surface measurements. On the other hand, neutron diffraction, thanks to a low attenuation of neutrons in most materials, is suitable for strain scanning of bulk samples. The conventional neutron diffraction method is based on the precise determination of the relative change in the *d_hkl_-*spacing of particularly oriented crystalline grains in the gauge volume [1,2,3]. In neutron and X-ray diffraction, the angular positions of the diffraction lines are determined by the well-known Bragg condition 2*dhkl* ⋅sin *θhkl* = *λ* (*θhkl*—Bragg angle, *λ*—the wavelength). The elastic strain *ε* is defined as *ε* = *Δd_hkl_/d_0,hkl_*. It is in fact the relative change in the lattice spacing with respect to the value *d_0,hkl_* corresponding to a strain-free sample element. The strain in the material is a tensor, and for its determination, at least three strain components should be obtained. However, only one component parallel to the scattering vector ***Q*** is determined from the measurement of one diffraction line (see the inset in Figure 1). Therefore, the strain measurements have to be carried out for three geometrical positions. It is clear that for strain evaluation, the determination of the *d_0,hkl_* value also represents a key task [1,2,3]. The differentiation of the Bragg condition provides a simple formula for the strain *ε* as *ε* = −cot *θhkl* ⋅*Δθhkl*. According to this formula, the strain *ε* in the material brings about a change in the scattering angle 2*θhkl* by *Δ*(2*θhkl*) of the position of the diffraction line with respect to the position of the line of the strain-free element. Therefore, from the angular shift in the diffraction line *Δθhkl*, the average lattice macrostrain component over the irradiated gauge volume element can be determined. The neutron strain/stress instrument is a 2-axis diffractometer (see Figure 1) equipped with a position-sensitive detector (PSD), usually used for the mapping of residual strains inside polycrystalline materials. The spatial resolution mainly determined by the beam defining optics is usually of the order of millimeters. The diffractometer is usually equipped with a focusing monochromator (bent perfect crystal) that has the curvature optimized to achieve a maximum luminosity and resolution with respect to a chosen diffraction line [1,2,3]. Together with the focusing monochromator, the beam optics system includes slits before and after the sample that determine the dimensions of the gauge volume element and, of course, the position-sensitive detector for imaging the beam profile diffracted by the irradiated element [4,5,6]. The strain/stress scanner can also be equipped with an auxiliary machine, e.g., for thermo-mechanical manipulation with the sample. Then, the resolution of the instrument results from the combination of uncertainties coming from the thickness and radius of curvature of the bent crystal monochromator, the widths of the slits (in most cases of 0.5–2 mm), the divergence of the incoming and diffracted beam by the irradiated element, and the spatial resolution of the position-sensitive detector. Their combination results in a total uncertainty usually forming a Gaussian image in the detector with a *FWHM* of about (5–10) × 10^−3^ rad, which can be sufficient for the measurement of the angular shifts of diffraction lines generated by strains. However, such resolution is not sufficient for the investigation of the changes in the diffraction profile, which would, e.g., allow for microstructure characterization of plastically deformed polycrystalline samples.

Encouraged by the first recent preliminary results [7], we have decided to study resolution properties of a new unconventional high-resolution neutron-diffraction three-axis setup for large bulk polycrystalline samples experimentally with the aim of a possible exploitation in the field of strain/stress measurements. The most important obtained results are presented.

## 2. Materials and Methods

Several years ago [8,9,10], the first attempts were made with a high-resolution three-axis setting, as schematically shown in Figure 2a. Following the drawing shown in Figure 2a (for small widths of the samples), the best resolution from this setting can be obtained when minimizing dispersion of the whole system. When solving the problem in momentum space, it means that the orientation of the *Δk* momentum elements related to the monochromator and analyzer should be matched to that of the sample, and this can be achieved by proper radii of curvatures of the bent perfect crystal monochromator and analyzer. Contrary to the conventional two-axis strain/stress scanner, the diffraction profiles with the three-axis setup are obtained by rocking the bent perfect crystal (BPC) analyzer situated on the third axis of the diffractometer, and the neutron signal is registered by the point detector (see Figure 2) [7,8]. By properly adjusting the curvature of the analyzer, the three-axis setting exploits the focusing in both real and momentum space. The resolution of this alternative setting was optimized by using a well-annealed α-Fe low-carbon steel rod (technically pure iron with C less than 1%) of the diameter of 8 mm (standard sample). For the experimental studies, the three-axis neutron optics diffractometer installed at the Řež research reactor LVR-15 (Research Centre Řež, Husinec, Czech Republic) and equipped with Si(111)-monochromator and Ge(311)-analyzer single-crystal slabs of dimensions 200 × 40 × 4 and 20 × 40 × 1.3 mm^3^ (length × width × thickness), respectively, was used. The Si(111) monochromator providing the neutron wavelength of 0.162 nm had a fixed curvature with the radius *R*_M_ of about 12 m, and the radius of curvature of the analyzer was changeable in the range from 3.6 to 36 m. Figure 3 shows some results of the optimization procedure related to the setting shown in Figure 2a, when searching for a minimum *FWHM* of the rocking curve, and the optimum curvature of the analyzer. The optimum radii of curvature of the Ge(311) analyzer for α-Fe(110) and α-Fe(211) reflections have been found as *R*_A_ = 9 m and *R*_A_ = 3.6 m, respectively. However, it has been found that high-resolution determination of the lattice changes can be achieved even on large irradiated gauge volumes (see Figure 2b), though slightly relaxed.

## 3. Results

After the adjustment of the diffractometer setting for optimum resolution, the first step was measuring the resolution of the experimental setting for different diameters of the well-annealed α-Fe(110) standard samples situated in the vertical position (see Figure 4). Concerning the resolution represented by the *FWHM* of the analyzer rocking curves, it can be seen from Figure 4 that the diameter of the sample (generally the thickness) plays an important role. In the next two steps, the influence of the slit width on the resolution properties for two diameters of the well-annealed sample of 4.9 and 8 mm in the horizontal position was studied (see Figure 5 and Figure 6). In both cases, three slit widths of 5, 10, and 20 mm were used. This can be useful when comparing the effect of different widths of the irradiated sample, as the vertical dimension of the irradiated sample does not play a principal role with respect to the resolution of the experimental setting. As expected, it can be seen from Figure 5 and Figure 6 that the irradiated width of the sample (represented in our case by the slit width) plays an important role. However, for the slit widths of 5 and 10 mm, the *FWHM* is also slightly influenced by the diameter of the sample. The inspection of Figure 4, Figure 5 and Figure 6 reveals that with the exception of the slit width of 20 mm, in all other cases, the angular resolution was sufficiently high for observation of not only elastic strains, but also changes in the diffraction profiles for microstructural (microstrains, mean grain size) studies of plastically deformed samples on the basis of diffraction profile analysis [11,12]. Furthermore, the comparison of Figure 5c with Figure 6c shows that in the case when the slit width is much larger than the diameter of the sample, the diameter itself has a negligible influence on the *FWHM* of the rocking curve. Finally, a slightly deformed α-Fe(110) low-carbon steel rod of *φ* = 4.9 mm put in the horizontal position for two slit widths was measured. The corresponding experimental rocking curves are shown in Figure 7. It should be pointed out that for measurements related to Figure 4b and Figure 5, the same sample was used, however, after annealing. If we compare Figure 7a,b with Figure 5a,b, the effect of plastic deformation on the *FWHM* of the rocking curve, as well as the diffraction profile itself, is evident and could be sufficient for a possible evaluation of the microstructure parameters on the basis of the diffraction profile analysis. Of course, if the samples are situated at the diffractometer with a high accuracy, the relative change in the lattice constant *ε = Δd*_S_/*d*_0_ (strain) can also be derived from the peak position shift *Δθ*_A_. The peak shift *Δθ*_A_ corresponds to the change in the scattering angle *Δ*(2*θ*_S_) as *Δθ*_A_ = −*Δ*(2*θ*_S_). Then, with the help of the differentiated form of the Bragg condition *Δd*_S_/*d*_0_ = −*Δθ*_S_ cot *θ*_S_ (*d*_0_ is the lattice spacing of the virgin sample), the average elastic strain values *ε*_R_ (radial component) and *ε*_L_ (longitudinal component) within the irradiated gauge volume could be evaluated.

## 4. Discussion

The obtained results shown in Figure 5, Figure 6 and Figure 7 prove that the presented diffraction method provides a sufficiently high angular resolution, represented by the *FWHM* of the diffraction profile, which allows for the possibility of studying some plastic deformation characteristics (root-mean-square microstrains, as well as the effective grain size), by applying shape analysis on the neutron diffraction peak profiles [11,12]. The advantage of this method permits the investigation of large volumes of polycrystalline samples (several cubic centimeters), thus providing average values of microstructure characteristics over the irradiated gauge volume, e.g., as a function of macroscopic strain loaded on the sample by an auxiliary instrument. Further application of this method can be found in strain/stress studies of textured samples with a large size of grains, as the conventional diffraction method requiring small irradiated gauge volume would be problematic. Naturally, it could also be used for strain scanning in the polycrystalline materials similarly to the conventional two-axis strain/stress scanners when evaluating strains in small gauge volumes (few cubic millimeters) from peak position shifts *Δθ*_S_. However, for such measurements, this method would not be practical, because the step-by-step rocking curve analysis would make it much more time-consuming. On the other hand, when using the sample with a width of about 10 mm (or more), the resolution of the conventional two-axis scanner would be so relaxed that it would not allow us to investigate the effect of elastic strains from the shifts in the diffraction profiles nor the effects of plastic deformation on their *FWHM* and the form. As can be seen in Figure 1, the width of the sample gauge volume introduces a decisive uncertainty to the resolution, i.e., to the *FWHM* of the diffraction profile imaged by the position-sensitive detector.

## 5. Conclusions

The presented high-resolution neutron diffraction method can be successfully used for the strain/stress measurements, namely, on bulk samples exposed to an external thermo-mechanical load, e.g., in a tension/compression rig and in an aging machine. The bulk samples in the form of a rod with a diameter of several millimeters can be investigated in the vertical, as well as horizontal, position. Similarly, the bars of a rectangular form could also be tested. In comparison to the conventional two-axis neutron diffraction strain/stress scanners, the three-axis alternative offers a considerably higher *Δd*/*d* resolution. However, the presented method, if used for conventional elastic strain/stress scanning, would be much more time-consuming. On the other hand, when using a large-volume sample accompanied with a considerably higher detector signal, this drawback can be partly eliminated. Nevertheless, the presented method could be useful, namely, at high-flux neutron sources where the experimental results can be obtained within a reasonable measurement time. It can be stated that the presented method using the three-axis neutron diffraction setting can offer further complementary information to that achieved by the other methods commonly used.

## Figures and Tables

**Figure 1 materials-13-05449-f001:**
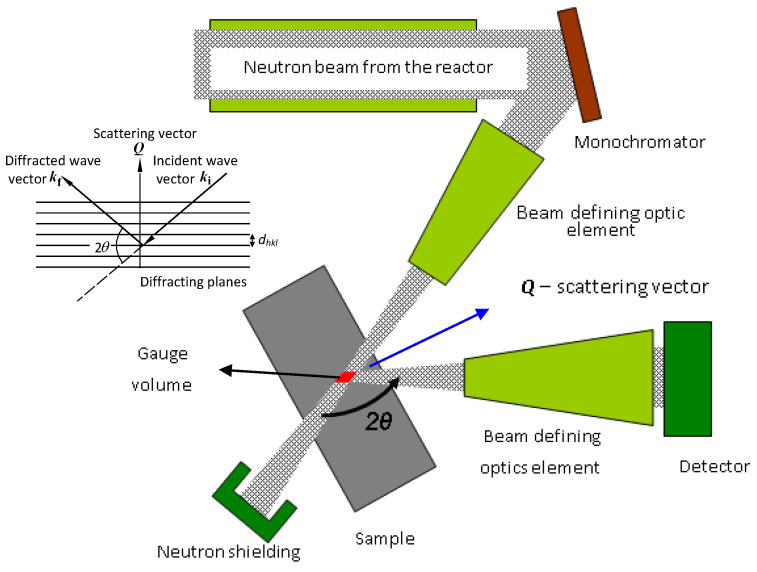
Scheme of the conventional strain scanner.

**Figure 2 materials-13-05449-f002:**
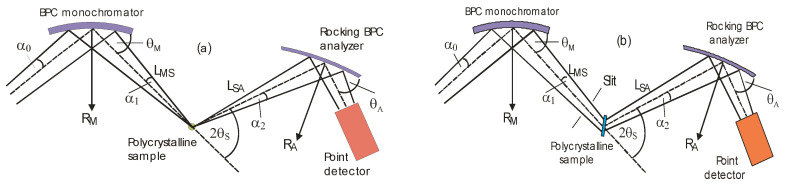
Three-axis diffractometer settings employing bent perfect crystal (BPC) monochromator and analyzer as used in the feasibility studies (*R*_M_, *R*_A_—radii of curvature, *θ*_M_, *θ*_A_—Bragg angles) for vertical (**a**) and horizontal (**b**) positions of a polycrystalline sample. This figure has been reprinted from Ref. [7].

**Figure 3 materials-13-05449-f003:**
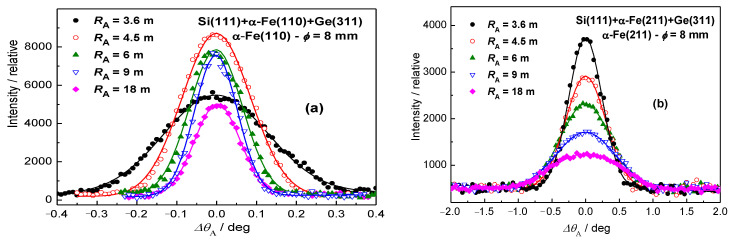
Analyzer rocking curves for different radii of curvature and for (**a**) α-Fe(110) and (**b**) α-Fe(211) reflections.

**Figure 4 materials-13-05449-f004:**
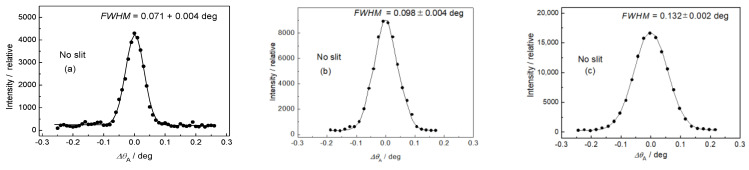
Rocking curves for the annealed α-Fe(110) low-carbon steel rods in vertical position: (**a**) *φ* = 2, (**b**) 4.9, and (**c**) 8 mm.

**Figure 5 materials-13-05449-f005:**
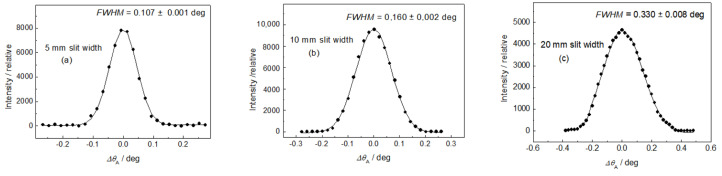
Rocking curves for the annealed α-Fe(110) low-carbon steel rods of *φ* = 4.9 mm in horizontal position: (**a**) 5, (**b**) 10, and (**c**) 20 mm Cd slits.

**Figure 6 materials-13-05449-f006:**
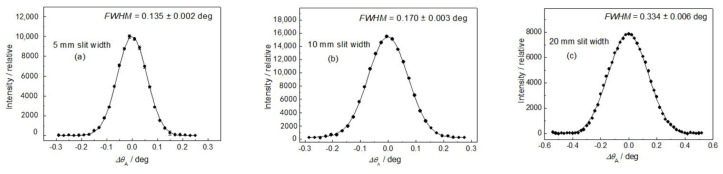
Rocking curves for the annealed α-Fe(110) low-carbon steel rod of *φ* = 8 mm in horizontal position: (**a**) 5, (**b**) 10, and (**c**) 20 mm Cd slits.

**Figure 7 materials-13-05449-f007:**
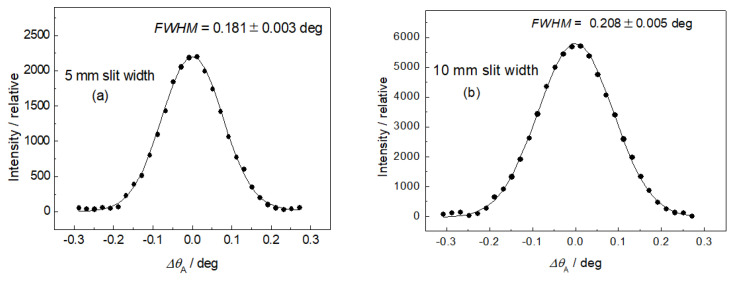
Rocking curves for the slightly deformed α-Fe(110) low-carbon steel rod of *φ* = 4.9 mm installed in horizontal position for (**a**) 5 mm slit width and (**b**) 10 mm slit width.

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
