# Peer review of "High-Resolution Strain/Stress Measurements by Three-Axis Neutron Diffractometer"

_materials, 2020, doi:10.3390/ma13235449_

Round 1

Reviewer 1 Report

Comments in attached file.

Author Response

R: P2 L37: Strain in a solid continuum is a (2nd-order) tensor, not a vector

A: It will be corrected in the paper

R: P4 L101-104:

A:

a) First attempts were already carried out (see [8-10])

b) The difference in FWHM for virgin and deformed sample cannot be an artefact, because the measurements were repeated on the used and several other  samples.

c) The peak broadening on bulk samples cannot be studied on an equivalent 2-axis machine because the width of the gauge volume introduces FWHM of the diffraction profile an unacceptable large uncertainty. As to the synchrotron radiation, I am not familiar  with the measurements of bulk samples on a three axis setting.

P5 L134: ....a severe impact on measurement time.

A: From the principle the presented method is much more time consuming if used for conventional strain/stress scanning. However, the detector signal is roughly proportional to the gauge volume of the sample and therefore, the measurement of bulk samples considerably decreases this drawback. According to our estimation the measurement time of one rocking curve with a sufficiently good statistics on a bulk sample at a high flux neutron source would be maximum several minutes. Of course that at at a rather lower flux neutron source (which is our case) the measurement time is considerably larger. This point will be reflected in the paper

Reviewer 2 Report

Reviewer comments

The manuscript stated on “On possible high-resolution residual strain/stress measurements by three-axis neutron diffractometer”

It must be revised before publishing.

  • The abstract part is so simple. It should be improved.
  • The introduction part has been stated well.. However, the authors have used the citation [1-3] many times. such as on line. 26, line. 33, line 41. line 52. Please avoid this. More citations are needed. As well, you should move Figure 1 and Figure 2 to another section.
  • More real photograph related to your experiment should be added.
  • The image of three-axis neutron diffractometer set up should be added.

Author Response

Thank you very much for comments.

The abstract has been improved.

The citation [1-3] was omitted in one case

We can use in the paper only scheme of the diffractometer. In our case it is not allowed to do photos in the reactor hall.

One citation has been added

Round 2

Reviewer 2 Report

Reviewer comments

Thank you so much for your revised version.

The quality of paper has been improved.

  • I suggest to check your paper roughly again. Many errors in your paper.

For example: Why you use the strikethrough on your image in Figure 1. Is it the removed image? Please delete all the strikethrough, and highlight your new version in red color.

  • Again, why you use the strikethrough for Figure 3, Figure 4, Figure 5.
  • As well on section “2. Materials and Methods” the description about your material is not clear…If possible, please add the material properties and chemical compositions of the material to your paper.

It is the "scientific research" not the “report”…Try to improve your quality of paper before it can publish.

Author Response

Answers to reviewer comments

A: Thank you very much for constructive comments

Abstract

A: Instead of „New unconventional…“, we prefer „Resolution properties of the unconventional …“

Introduction

A: Originally, we submitted the paper with two schematic figures 1 and 2 which were already printed in ref. 7 (Powder Diffraction). In order to avoid the reprinting, in the revised version we have used another schematic figure with the inset and thus have used only one Figure 1 instead of two figures in the originally submitted paper. 

A: We do not fully understand, what the reviewer means with the expression (requirement) “strikethrough”. If it means comparison of originally submitted paper with the revised one, we have the following explanation:  

 In the originally submitted paper together with other figures we used several reprinted figures, namely, schematic drawings Figure 1, Figure 2, Figure 3 and then two experimental rocking curves, namely, Figure 4a and Figure 5. As we had at our disposal new recent experimental results, in the revised version of the paper we reprinted only the schematic drawing of the experimental setting -Figure 3 (in the revised version it is Figure 2), omitted all other mentioned reprinted figures and used new ones which we think could better prove the goal of the publication.

Figures 4-7 in the revised paper correspond to the experimental results which have not been published before.  

  1. Materials and Methods

R: add the material properties and chemical compositions

A, lines 87-88: a-Fe low-carbon steel rod (technically pure iron with C less than 1 %)

Line 119: Instead of „industrial a-Fe(110) steel wire“ we use „a-Fe(110) low-carbon steel rod …“

Captions

Similarly, the samples are better described in the captions to Figures 4-7.

In the caption to Figure 2 the comment „This Figure 2 is reprinted from ref. [7]“ was added.

In the captions to Figures 4-7 „…the annealed a-Fe(110) low carbon steel rods…“ was added.

In the caption to Figure 7 „…low carbon steel rod…“ was added.

References

Two misprints in the references have been corrected.

We would like to ask reviewer to look at the pdf file. In Word file we do not see strikethrough figures.

On behalf of authors, Pavol Mikula